# Vitamin D Status in Belgian Children: A Regional Study

**DOI:** 10.3390/nu16050657

**Published:** 2024-02-26

**Authors:** Louise Van de Walle, Yvan Vandenplas, Jaan Toelen, Anke Raaijmakers

**Affiliations:** 1Department of Development and Regeneration, University of Leuven, 3000 Leuven, Belgiumjaan.toelen@uzleuven.be (J.T.); 2KidZ Health Castle, UZ Brussel, Vrije Universiteit Brussel (VUB), 1090 Brussels, Belgium; yvan.vandenplas@uzbrussel.be; 3Department of General Pediatrics, University Hospitals Leuven, 3000 Leuven, Belgium; 4Department of Paediatric Nephrology, Sydney Children’s Hospital Randwick, Sydney Children’s Hospital Network, Randwick, NSW 2031, Australia; 5Randwick Clinical Campus, School of Women’s and Children’s Health, University of New South Wales, Randwick, NSW 2031, Australia

**Keywords:** vitamin D, vitamin D status, vitamin D deficiency, rickets prevention

## Abstract

*Background:* Vitamin D deficiency is the most frequent cause of impaired skeletal growth, and can lead to the development of nutritional rickets. The aim of this study was to evaluate the vitamin D status in a large group of children aged 0–18 years. *Methods:* We collected laboratory data on vitamin D levels from children who underwent blood sampling between 2014 and 2021. *Results:* We included 14,887 samples. In this group, 17.7% were vitamin D severely deficient (<12 ng/mL), 25.2% were insufficient (12–20 ng/mL), and another large proportion (28.3%) was borderline (20–30 ng/mL). Sufficient levels (>30 ng/mL) were met in 28.8% of children. We observed no association between gender and vitamin D status (*p* = 0.132). Adolescents aged 13–18 years (*n* = 3342) had the highest prevalence of severe vitamin D deficiency (24.9%). Vitamin D levels were higher in summer/autumn compared to winter/spring. *Conclusions:* Vitamin D deficiency/insufficiency has a high prevalence in children, mostly in children above 7 years of age. Many of these children (over 80%) do not meet the 30 ng/mL sufficiency threshold. It is essential that Belgian Health Authorities are aware of this high prevalence, as the current Belgian recommendation suggests ceasing vitamin D supplementation at the age of six. Additional research is required to investigate the consequences of our findings, and what specific approach is needed to achieve normal vitamin D levels in children aged 0 to 18 years.

## 1. Introduction

Vitamin D plays a pivotal role in calcium and phosphate homeostasis, as well as in healthy bone metabolism, especially during growth [1]. In addition to the role of vitamin D in growth, it is believed that it might also plays a role in the regulation of the immune system [2,3].

Vitamin D is an umbrella term which comprises vitamin D3 (cholecalciferol) and vitamin D2 (ergocalciferol). Both forms are found in nature; vitamin D2 is derived from plants, whereas vitamin D3 is found in animal sources. However, the amount of vitamin D in natural food sources is low, and the contribution of dietary vitamin D is small or even negligible [4,5]. The main source of vitamin D3 comes from 7-dehydrocholesterol (previtamin D3) synthesis in the skin, in response to UVB rays. Solar UVB intensity is based on the season (higher radiation in summer) and geographical latitude (higher radiation in areas closer to the equator). The cutaneous synthesis of previtamin D3 cannot be detected above 50° geographical latitude from October to March [6], hence people in these areas (including Belgium) are at risk of low vitamin D levels, particularly in winter.

There is much debate about the optimal concentration of vitamin D in children. Research on pediatric vitamin D references is scarce [6]. Most studies on the optimal vitamin D levels are performed in adults [7]. Most experts agree that levels <12 ng/mL (30 nmol/L) represent severe deficiency, as these levels carry a high risk of rickets, even with an adequate calcium intake; today, most experts use 20–30 ng/mL (50 nmol/L) as the target value, which includes a safety margin [1,2,8,9]. For levels >30 ng/mL (75 nmol/L), the concentration of parathyroid hormones remains constant, which means that there is no secondary hyperparathyroidism. Some authors state that >30 ng/mL would be the logical ‘optimal’ vitamin D level [10], which is supported by the European Society of Pediatric Nephrology [9].

Vitamin D deficiency is the most frequent cause of impaired skeletal growth. Moreover, defective bone mineralization can lead to the development of rickets (or osteomalacia in older children) [11,12]. Globally, nutritional rickets, due to insufficient vitamin D and/or calcium, is the leading cause of pediatric bone disease [11,12]. However, the diagnosis of rickets is challenging, as the symptoms are insidious and difficult to recognize [8]. The use of surrogate markers is usually recommended [9]. Secondly, clinical research has demonstrated that vitamin D deficiency is associated with many acute and chronic illnesses, including calcium metabolism, several autoimmune diseases and cancers, type 2 diabetes mellitus, cardiovascular diseases, and potentially many more [4,7,13].

More than one billion humans worldwide are affected by vitamin D deficiency and insufficiency [1]. Additionally, a recent study performed in Belgium confirmed that vitamin D insufficiency and deficiency is still common in children, particularly at the end of winter [14]. In this study, we investigated the vitamin D status in children, visiting a regional hospital network (ZNA) in Antwerp, Belgium. We assessed vitamin D statuses and the factors associated with a poor status in a large group of children (0–18 years).

## 2. Materials and Methods

### 2.1. Data Collection and Subjects

Vitamin D data were retrospectively collected from the hospital central laboratory system, and personal data, barring sex and date of birth, were not including in the dataset. Children (aged 0–17 years and 364 days) who underwent a blood test including a vitamin D level in the ZNA hospitals network between 2014–2021 (*n* = 26,233) were retrieved. These years were chosen deliberately, as they represent the period after the consensus of vitamin D supplementation by the Flemish Society of Pediatrics [15]. Additional demographics (age, gender, and date of sampling) were also extracted. Individual data on vitamin D supplementation, diet, and level of sun exposure were not available, due to the retrospective and anonymous nature of the study.

We only included first-time samples in every patient, as subsequent samples are likely to be biased by comorbidities (unwell or chronically ill patients will have more samples) and/or treatment (low levels will trigger treatment). Samples taken at birth were considered maternal values and were excluded. Children having multiple (≥2) vitamin D values measured in their sample or with missing information regarding their date of birth or gender were also excluded. The final sample size was *n* = 14,887 (Figure 1).

### 2.2. Laboratory Assessment and Reference Values

The quantitative determination of total 25-OH-vitamin D in the serum was performed by electrochemiluminescence binding assays (ECLIA) using Cobas E immunoassay analyzers (Roche Diagnostics, Mannheim). Vitamin D cut-offs were considered as per most used definitions of severely deficient, insufficient, borderline, and sufficient vitamin D levels of <12 ng/mL, 12–20 ng/mL, 20–30 ng/mL, and >30 ng/mL, respectively (Table 1) [1,2,8,9].

### 2.3. Statistical Analysis

For database management and statistical analysis, we used SPSS Statistics for Macintosh (Version 28.0, Armonk, NY, USA). Data were measured at the ratio level. No significant outliers were identified based on the visual inspection of the created boxplots.

Frequency tables were calculated based on vitamin D levels in relation to age group and season. Differences in the frequency distributions of vitamin D status between categories were assessed using Pearson’s Chi-Square test. Significant differences were considered if the *p*-value was <0.05. The following age categories were defined: <1 year, 1–6 years, 7–12 years, and 13–18 years. Seasons were defined according to the following months: spring (March, April, and May), summer (June, July, and August), autumn (September, October, and November) and winter (December, January, and February).

## 3. Results

### 3.1. Characteristics of the Cohort

Baseline characteristics are provided in Table 1. Males were slightly overrepresented in the youngest age group, while females were more prominent in the oldest group. The largest group were those aged between 1 and 6 years old (*n* = 5070, 34.1% of total sample size). Children up to 1 year old were the smallest group (*n* = 2354, 15.8% of total sample size).

### 3.2. Vitamin D Status

In our total sample, the distribution of the vitamin D serum concentration had a mean value of 23.94 ng/mL and a standard deviation of 12.11 ng/mL. The lowest level of concentration was 3.04 ng/mL. The highest level was 58.41 ng/mL.

Table 2 contains the distribution of vitamin D statuses in both absolute numbers and percentages within each age group for the total sample. Overall, 2635 (17.7%) were severely deficient, 3746 children (25.2%) were insufficient (12–20 ng/mL), and 4215 children (28.3%) were borderline (20–30 ng/mL). In total, 71.2% of the population did not meet the >30 ng/mL threshold, most prominent in the over-7 age groups (83.6% and 83.0% of children in the 7–12 and 13–18 age groups, respectively). There was no significant association between the vitamin D status and gender (Pearson Chi-Square: *p* = 0.132).

The prevalence of vitamin D insufficiency along with vitamin D deficiency was highest in 7–12-year-old children and adolescents (13–18 years), being 24.9% and 20.7%, respectively. In the youngest age group, 23.2% of the children had insufficient vitamin D levels, and 22.3% had severely deficient vitamin D levels. Overall, the groups of children above 7 years of age had the lowest vitamin D status.

In our study population, we were vigilant of seasonal variation. Vitamin D values measured during summer and autumn were significantly higher when compared to values measured in winter and spring. The proportion of vitamin D-sufficient children were highest in the summer and autumn months, as shown in Figure 2.

Figure 2 is a graph showing the percentages of children who were vitamin D severely deficient, insufficient, borderline, and sufficient, according to the season. The percentages shown in black, dark grey, white, and light grey represent levels <12 ng/mL, 12–20 ng/mL, 20–30 ng/mL, and >30 ng/mL.

## 4. Discussion

Vitamin D deficiency is prevalent among Belgian children, with the highest prevalence in adolescents. The proportion of children with severely deficient vitamin D levels rises significantly with age, and reaches a peak before adolescence, where up to 55% of children have vitamin D levels below 20 ng/mL, and up to 84% have a vitamin D level that does not meet the sufficiency threshold (i.e., above 30 ng/mL). Additionally, our data confirm that at the end of winter vitamin D status is the lowest, when up to 77% of children have vitamin D levels below 30 ng/mL.

Our research shows results in line with studies in other European countries [6,16,17]. Furthermore, a recent study conducted in Belgium showed that up to half their population of adolescents have very low (<20 ng/mL) vitamin D levels [14]. We acknowledge that the cut-offs used in this study could be debated, and the stricter we would be, the more striking the results. The European Society for Pediatric Nephrology have developed recommendations for the evaluation, treatment, and prevention of vitamin D deficiency [9]. Their aim was to provide guidance on children with chronic kidney disease, but their systematic approach covered healthy children too. They recommend using levels above 30 ng/mL as sufficient [9], and our study concurred.

The reasons for the high prevalence of low vitamin D levels could be plenty. Firstly, (the lack of) supplements could play a role. The administration of 400 IU/day to babies during their first year of life, regardless of season, diet, or other potentially influencing factors, is a generally accepted recommendation in Europe to prevent nutritional rickets in babies. Despite this, our results show that only a third (32.3%) of the infants had sufficient vitamin D levels. These results are in line with previous studies, which showed that Belgian babies often have insufficient vitamin D levels [14,18].

Secondly, children with chronic diseases or conditions such as obesity do need extra vitamin D, especially in winter [1,3,7,9,19]. Vitamin D deficiency in obese children is mostly due to dilution in the larger volumes of fat, serum, liver, and muscle present in obese children [20]. According to a previous Belgian study, vitamin D levels are significantly influenced by body composition [21]. Similarly, an Italian study showed that children who were overweight had higher levels of vitamin D deficiency than those with optimal weight [22]. Unfortunately, we did not have data on body composition available for our study.

Thirdly, children with limited sun exposure are more likely to suffer from a vitamin D deficiency [5]. Although Italy (45.6–41.5°) is located below 50° geographical latitude, we detected the same high prevalence of vitamin D deficiency as detected in Antwerp, located at 51.2° latitude [17,23]. Sunlight is thought to be the most important source of vitamin D, but the same high prevalence of vitamin D deficiency in both countries might indicate that other factors, like differences in lifestyle (outdoor play, veiling, etc.), diet (including food-fortification policies, vegetarian or vegan diet, etc.), vitamin D supplementation, or/and socio-economic dissimilarities, are as important as sun exposure [23]. By achieving sufficient solar exposure, the debate on proper photoprotection (to protect children against sunburn, cancer and photoaging) and the importance of sunlight exposure to establish adequate vitamin D levels rises in the majority of healthcare professionals and parents [24]. The effect of sunscreen application on vitamin D synthesis remains a topic of debate [1,24]. However, it is likely that the combination of sunscreens, along with covered clothing, dark pigmentation of the skin, and shade-seeking behavior are more likely to compromise vitamin D status in children [24]. Our results could imply that children lack sufficient safe sun exposure. Moreover, a low socioeconomic status is a risk factor for vitamin D deficiency [25,26].

Potential interventions should be multifaced too [7]. Firstly, our results seem to suggest that the WHO recommendation (administer supplements to all infants <1 year) is either not well-communicated and/or followed-up by healthcare professionals (pediatricians, general practitioners, well baby clinics, etc.) or not well administered/followed by parents. Raising awareness through straightforward communication and parental education could improve vitamin D levels in neonates. However, infants absorb calcium independently of vitamin D [27,28], so the clinical consequences of low levels in infancy might be limited. Moreover, the recommendation in Belgium is to supplement 400 IU of vitamin D daily for all children aged up to the age of six [15]. Based on the high prevalence of hypovitaminosis in children aged over 6 in our study, we suggest that maintaining vitamin D supplementation beyond the age of 6 could reduce the risk for vitamin D deficiency at a later age.

Secondly, individualized administration of supplements in children based on the presence of risk factors for low vitamin D production and intake (e.g., obesity, malabsorption diseases, veiling, …) should always be considered in clinical practice.

Lastly, sun exposure is the major source of vitamin D [7]. Research shows that playing outside for at least 15 min per day with uncovered arms and legs between 10 am and 3 pm and in the months between May and October would be sufficient to have adequate vitamin D levels throughout winter for children with pale skin [22]. The importance of extending sun-safe outdoor time in childhood should be highlighted properly in schools by promoting playing outdoor during all seasons.

Lastly, besides recommending supplements and adequate sun exposure, along with photoprotection, making vitamin D supplements and/or wider food fortification with vitamin D could also help to meet the vitamin D requirements in children [7,29,30]. Based on our study, action is needed to achieve sufficient vitamin D levels, especially in adolescents.

This study should be interpreted within the context of its limitations. Firstly, this is an observational retrospective study that does not consider biometrics, sun exposure, supplementation, nutrition, or ethnicity. Due to the nature of our study, detailed information was not available. Additionally, the indication for the blood test was not available, and this potential bias should be considered when interpreting the results. Secondly, children visiting a hospital were used to select our study population. This population does not perfectly reflect the general population. Well-known risk factors (chronic kidney disease, chronic liver disease, use of certain drugs, malabsorption, cholestasis, etc.) were not accounted for. However, given the large sample size, we consider the study population to be a representative sample, keeping in mind that unwell children or children with risk factors are more likely to be sampled when compared to their ‘healthy’ counterparts. Also, clinicians ordering the blood test could have a clinical suspicion of vitamin D deficiency that was not accounted for in our study, and would therefore create an additional bias. Lastly, because of the absence of data regarding PTH, phosphate, and bone density, conclusions about bone strength and clinical consequences in children with vitamin D deficiency cannot be drawn.

Additional research is required to investigate the consequences of our findings. Identifying causes might be effective to prevent and treat vitamin D deficiency. Also, research on specific approaches is needed to achieve normal vitamin D levels in children aged 0 to 18 years, especially in adolescents.

## 5. Conclusions

We report a very high prevalence of low vitamin D levels during childhood, mostly during late childhood and adolescence, in a large retrospective analysis of laboratory data in children. A huge proportion (over 70%) of these children do not meet the 30 ng/mL sufficiency threshold, especially children above 7 years of age. Additionally, we confirmed a seasonal variation with the lowest vitamin D serum values in winter and spring. No association between vitamin D and gender was found. It is essential that national health authorities are aware of the high prevalence of vitamin D deficiency in children, as the current Belgian recommendation would imply ceasing vitamin D supplementation at the age of six.

## Figures and Tables

**Figure 1 nutrients-16-00657-f001:**
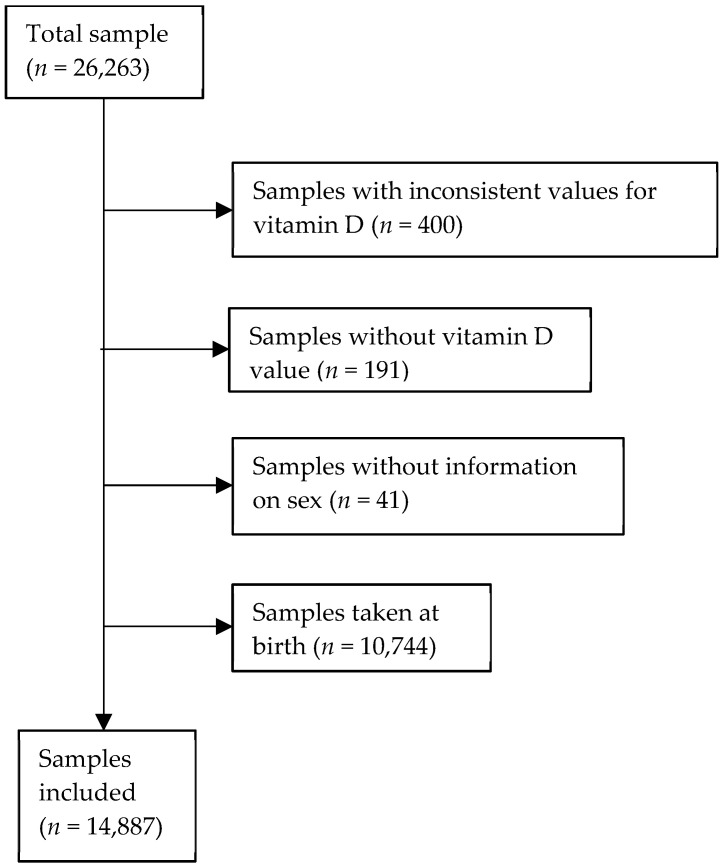
Overview of the total sample and the excluded and final included samples.

**Figure 2 nutrients-16-00657-f002:**
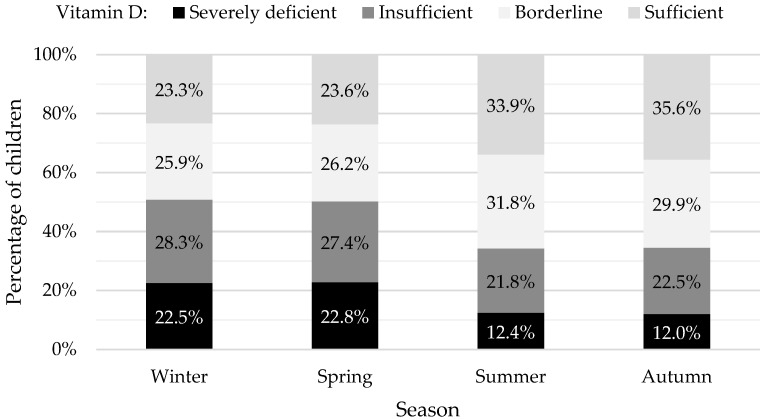
Percentages of vitamin D sufficient, insufficient, and deficient children.

**Table 1 nutrients-16-00657-t001:** Sample sizes by age and gender.

		Total (*n*)	Female (*n*)	Proportion (%)	Male (*n*)	Proportion (%)
Age in years	<1	2354 (15.8%)	1079	45.8%	1275	54.2%
	1–6	5070 (34.1%)	2348	46.3%	2722	53.7%
	7–12	4121 (27.7%)	2141	52.0%	1980	48.0%
	13–18	3342 (22.4%)	2016	60.3%	1326	39.7%
	Total	14,887 (100.0%)	7584	50.9%	7303	49.1%

**Table 2 nutrients-16-00657-t002:** Vitamin D status by age groups.

	Age						
			Years				Total
			<1	1–6	7–12	13–18	
Vitamin D status	Severely deficient <12 ng/mL	Count (*n*)	526	423	853	833	2635
		Percentage	22.3%	8.3%	20.7%	24.9%	17.7%
	Insufficient 12–20 ng/mL	Count (*n*)	547	875	1343	981	3746
		Percentage	23.2%	17.3%	32.6%	29.4%	25.2%
	Borderline 20–30 ng/mL	Count (*n*)	521	1483	1250	961	4215
		Percentage	22.1%	29.3%	30.3%	28.8%	28.3%
	Sufficient >30 ng/mL	Count (*n*)	760	2289	675	567	4291
		Percentage	32.3%	45.1%	16.4%	17.0%	28.8%
	Total	Count (*n*)	2354	5070	4121	3342	14,887

## Data Availability

Data are available upon reasonable request.

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
