# Peer review of "Vitamin D Status in Belgian Children: A Regional Study"

_nutrients, 2024, doi:10.3390/nu16050657_

Round 1

Reviewer 1 Report

Comments and Suggestions for Authors

In the manuscript entitled "Vitamin D status in Belgian children: a regional study”  authors provide the valuable information on the vitamin D status in a large cohort of Belgian children. Although there is a lot of lacking data, mentioned by authors in lines 200-213, I consider this data of great value for any populational studies and meta-analyses in this topic. 

However, my major concern is for the methodological approach dividing values for deficiency below 12 ng/ml, insufficiency as the concentration of 12-20 ng/ml, and sufficiency >20 ng/ml. First, in line 82 the authors cited three publications. In the paper cited as #1- Holick et al. did not mention such division, however, several times Holick et al pointed out, that the concentration below 20 ng/ml should be considered a deficiency, the level of 21-29 ng/ml- as the insufficiency, and the level of 25(OH)D3 above 30 ng/ml should be claimed as the vitamin D sufficiency (additional citation of Hollick et al from 2011 published in Lancet). 

The second mentioned citation in line 82 is the paper by Antonucci et al indeed mentioned such division, as one of the possible solutions for the definition of vitamin D deficiency/sufficiency (particularly the paper presenting such division is: DOI:10.1210/jc.2015-2175), however here authors also state that: "In contrast, other scientific societies have set the cutoff level for vitamin D sufficiency at ≥30 ng/mL".

The last cited paper published in Lancet in 2003 states only that the value of 10 ng/ml (25 mmol/L) in children is deficient and above this threshold, the values are normal (however this problem is not well discussed in this seminar paper).

The authors should reconsider using different cutoff levels in presenting their data. The level of 30ng/ml, also mentioned by the authors as critical, should be used.

Additionally, in the introduction more information about the pleiotropic effect of vitamin D might give a better overview of the subject.

Reviewer 2 Report

Comments and Suggestions for Authors

In this manuscript the authors report the vitamin D status of children in a specific area of Belgium.

This is an interesting and well written manuscript.

The main problem with the manuscript, and the authors briefly discuss it at the end, is the patient selection. This is not a study where all the blood samples of children were analyzed. In this study there was an “original” request to assess Vitamin D levels in the blood specimen. This is an important bias in sample collection since, in theory, the health care professional requesting the exam already had a suspicion of inadequate vitamin D levels.

The authors should go into more details about this in the limitations section.

Round 2

Reviewer 1 Report

Comments and Suggestions for Authors

The authors introduced this statement into the manuscript: "Our research shows results in line with studies in other European countries [7,16–18]. Furthermore, a recent study conducted in Belgium showed that up to half their population of adolescents have low (<20 ng/ml) vitamin D levels [14]. We acknowledge that the cut-offs used in this study could be debated and the stricter we would be, the more striking the results. If we would use levels >20 ng/ml as sufficient, almost half (42.9%) of our population would be deficient with ~50% of children not meeting ‘adequate’ levels in winter and spring" . 

This information is misleading and highly inaccurate. The cut-off of 30 ng/ml will be of great interest to any researcher using this data for meta-analysis. Indeed, 40 ng/ml would also have a great value for some researchers.

The 20 ng/ml might still be called a "sufficient" level- if the authors insist upon this value. However,  the information of higher values would be as important as the deficiency levels. Not including higher cut-offs in this study, where the large number of participants is the main advantage, would drastically decrease the value of this publication.
